# OpenReview forum: "The Perplexity Paradox: Why Code Compresses Better Than Math in LLM Prompts"
_TMLR — Under review for TMLR_

### Review · Reviewer_yjce · 2026-04-21

**Summary Of Contributions:**

This paper studies prompt compression for LLMs through the lens of cost-quality trade-offs across task types. It reports that code-generation prompts exhibit a threshold-like tolerance to compression, whereas chain-of-thought reasoning degrades more gradually, and it controls for prompt-length confounds via ANCOVA and length-matched sampling. To explain these differences, the authors present a per-token analysis that motivates a perplexity paradox. Building on these insights, the paper proposes TAAC, a task-aware adaptive compression strategy that dynamically selects compression ratios using task classification and quality-gated stopping. Experiments across code and reasoning benchmarks, together with targeted interventions such as signature preservation, aim to support both the empirical patterns and the proposed mechanism.

**Audience:**

No

**Audience Explanation:**

The topic is interesting for readers but the paper is not solid.

**Claims And Evidence:**

No

**Claims Explanation:**

There are many problems needed to solve:

1. The key reference [1] is not accessible for me. It looks a fake reference.

2. The paper frequently states code performance as being largely preserved above a compression ratio around r≥0.6 (e.g. see Fig.1 caption). However, the statement is not true, since reported code results (MBPP pass rate and quality retention) still show a noticeable drop at r=0.6 compared with the uncompressed baseline (see Table 5 and Table 14) . I suggest the authors to revise the statement carefully.

3. The paper motivates a “threshold homogeneity” hypothesis across compressors, but parts of the multi-compressor section read as planned or expected rather than fully demonstrated in the main results. This makes it harder to assess whether the central pattern holds broadly across compression algorithms and model families (see Section 6.2~6.4).  Clarify this point is recommended.

4. The introduction and abstract describe broad coverage across many benchmarks and languages, while the main quantitative narrative is most detailed for a subset (notably MBPP and the length-controlled analyses), which can not provide enough evidence for the claimed research scope. (see Abstract and Section 1 against Section 8).

5. The paper alternates continuously between quality score, pass rates, and quality retention, which can make cross-task comparisons (code vs. reasoning) difficult to interpret. It is not always clear how these metrics relate, or whether they are intended to be directly comparable across tasks. (Section 3, 4.1, 4.4, 8)

6. The experimental setup would be easier to reproduce and audit with more complete configuration information in the main text. Please provide more details about all experimental configurations.

7. The presentation is quite bad and hard to read. The paper seems to have been generated by LLMs. Please clarify the usage of LLMs in producing the manuscript.

References:
[1] Compress or Route? Task-Dependent Strategies for Cost-Efficient Large Language Model Inference.

**Requested Changes:**

See the problems above.

---

### Review · Reviewer_Xj1g · 2026-04-23

**Summary Of Contributions:**

Double-blind problem

Hello TMLR,

I want to flag a possible double-blind concern in this submission.

The paper cites an anonymous 2026 work, but does so using self-identifying phrasing such as “In "Compress or Route?" (Anonymous, 2026), we found”, “the first paper of this research series”, and “our prior work”, rather than treating it as neutral anonymous related work.

While the "Compress or Route" paper must be read by the Reviewer to get the background of the paper, doing such self-identification immediately reveals the identity of authors. This breaches double-blind practice.

I would suggest revising the paper before resubmission.

Best,

**Additional Comments:**

N/A

**Audience:**

No

**Audience Explanation:**

N/A

**Claims And Evidence:**

No

**Claims Explanation:**

N/A

**Requested Changes:**

N/A

---

### Review · Reviewer_hqKJ · 2026-06-09

**Summary Of Contributions:**

Summary:
This paper investigates task-dependent prompt compression for large language models and proposes Task-Aware Adaptive Compression (TAAC), a quality-gated framework designed to reduce inference cost while preserving task performance. Motivated by different compression behaviors in code generation and chain-of-thought reasoning, the authors show through length-controlled experiments that these differences cannot be explained by prompt length alone. They introduce the “perplexity paradox,” arguing that compression methods often preserve high-perplexity code syntax tokens while discarding low-perplexity numerical values that are crucial for mathematical reasoning. Token-level perplexity analysis and signature-preservation experiments support this claim, showing that restoring function signatures after aggressive compression improves code pass rates and reduces NameError failures. TAAC addresses these issues by classifying prompts as code, reasoning, or hybrid, estimating information density with perplexity statistics, and applying iterative compression controlled by a quality predictor. The method stops compression when predicted quality drops below a threshold, outperforming fixed-ratio compression strategies. However, I have some concerns about this paper. My detailed comments are as follows.


Strengths:
1. A distinctive aspect of the paper is its attempt to provide a mechanistic explanation for task-dependent compression behavior through the “perplexity paradox”, which connects token-level perplexity patterns with the different robustness of code and reasoning prompts under compression.
2. The authors propose TAAC, a task-aware adaptive prompt compression framework that combines task classification, information-density estimation, and quality-gated compression to dynamically choose compression levels for different types of prompts.
3. The paper evaluates its claims from multiple complementary angles, including length-controlled analysis, MBPP validation, token-level perplexity analysis, signature-preservation experiments, and a comparison between TAAC and fixed-ratio compression baselines.

Weaknesses:
1. The main threshold claim is not well supported by the MBPP results. The reported MBPP performance shows continuous degradation rather than a clear threshold around 0.6, and the authors should either weaken this claim or provide stronger threshold analysis.
2. The novelty of the “perplexity paradox” is overstated. Prior work, such as LLMLingua-2 [A], and TACO-RL [B] has already discussed the mismatch between entropy/perplexity-based importance and downstream task utility, so the authors should clarify the specific novelty of their code-vs-reasoning analysis.
3. The mechanism analysis does not fully connect token perplexity, compression decisions, and downstream failures. Although the paper reports that Python syntax tokens have high perplexity and numerical tokens have relatively low perplexity, it does not sufficiently show that task-critical numerical values are actually pruned more often, nor that their removal directly causes reasoning failures. The authors should provide more direct evidence that task-critical numbers are pruned and that their removal causes reasoning errors.
4. The signature-preservation experiment is compelling but does not by itself validate the full perplexity-paradox mechanism. This experiment shows that restoring function signatures improves code-generation performance and reduces NameError failures, but it does not isolate whether the improvement is due specifically to perplexity-based preservation, function identity restoration or other structural constraints.
5. TAAC appears to be a heuristic combination of existing task-aware and quality-aware compression ideas. The authors should clarify the methodological novelty of TAAC and compare it directly with stronger task-aware baselines such as LLMLingua-2 [A], TACO-RL [B], and LLM-DCP [C].
6. The motivation for TAAC’s task-specific thresholds is not fully justified by the empirical results. In particular, the code threshold r* = 0.65 is motivated by the claimed r = 0.6, but the MBPP results do not show strong quality preservation at r = 0.6. The authors should provide a principled threshold-selection procedure, validation curves, and sensitivity analyses for the chosen thresholds for code, CoT, and hybrid tasks.
7. The information-density estimator in TAAC is under-motivated. The paper should empirically show that the coefficient of variation of token perplexity actually predicts compressibility or quality degradation.
8. The multi-algorithm validation is incomplete in the current manuscript. The paper presents a planned comparison among LLMLingua-2, LLMLingua-1, Selective Context, and Random compression, but the results section does not report the actual outcomes of this validation. Since one of the paper’s claims is that the task-dependent compression pattern is not an artifact of a particular compression method, the authors should include full multi-algorithm results or remove this claim.
9. Several experimental descriptions and reported numbers are inconsistent across sections. For example, the MBPP experiment is described as involving 9,000 trials, while the reported results use 1,800 trials; the length-matched analysis is described with slightly different sample sizes; and some effect sizes differ between earlier sections and the Results section. The authors should carefully reconcile these inconsistencies and provide a clear experimental accounting.
10. The TAAC evaluation is limited in scale and does not sufficiently establish generalization. The reported TAAC comparison uses a small synthetic validation set of 220 prompts, which is not enough to demonstrate robust performance across diverse real-world compression scenarios. The authors should evaluate TAAC on larger and more realistic workloads.
11. The ablation and diagnostic analysis of TAAC is insufficient. The authors should report component-wise ablations, quality-predictor calibration, task-classifier accuracy, and latency or cost overhead.
12. Some claims in the abstract and introduction appear stronger than what the experiments actually demonstrate. The paper claims broad validation across multiple code and reasoning benchmarks and languages, but the detailed reported results focus mainly on MBPP, length-controlled analysis, perplexity analysis, signature preservation, and TAAC evaluation. The authors should either provide complete results for all claimed benchmarks or revise the abstract and introduction to more accurately reflect the evidence presented.

*Reference:*
[A] LLMLingua-2: Data Distillation for Efficient and Faithful Task-Agnostic Prompt Compression. ACL 2024.
[B] TACO-RL: Task Aware Prompt Compression Optimization with Reinforcement Learning. ACL 2025.
[C] Dynamic Compressing Prompts for Efficient Inference of Large Language Models. TKDE 2026.

**Audience:**

Yes

**Audience Explanation:**

The paper studies a relevant and practical question: whether different task types tolerate prompt compression differently and why compression may affect code and reasoning prompts in different ways. The observations about task-dependent compression behavior, function-signature preservation, and the possible mismatch between token perplexity and task utility are potentially useful to researchers designing compression or routing systems. However, this interest is conditional on the authors clarifying and narrowing several claims, since the current evidence does not convincingly support the strongest conclusions about a robust compression threshold, multi-algorithm generality, or the full perplexity-paradox mechanism.

**Claims And Evidence:**

No

**Claims Explanation:**

The submission contains several interesting observations, but its central claims are not yet supported by sufficiently accurate, convincing, and clearly organized evidence. In particular, the claimed robust code-compression threshold around 0.6 is not convincingly demonstrated, since the MBPP results show continuous degradation rather than a clear threshold. The proposed “perplexity paradox” is plausible, but the evidence does not fully connect token perplexity to actual compression decisions and downstream failures, especially for task-critical numerical values in reasoning prompts. The TAAC method is also only weakly validated, with limited evaluation on a small synthetic set, insufficient ablations, and no direct comparison with stronger task-aware compression baselines. Moreover, the paper describes a multi-algorithm validation but does not report the corresponding results, and several reported experimental numbers are inconsistent across sections. Overall, there is a substantial gap between the paper’s main claims and the evidence currently presented, and the narrative would need either stronger experiments or significantly reduced claims.

**Requested Changes:**

See Weaknesses.

---

### Comment · Reviewer_Xj1g · 2026-04-21
**Double-blind problem**

Hello TMLR,

I want to flag a possible double-blind concern in this submission.

The paper cites an anonymous 2026 work, but does so using self-identifying phrasing such as “In "Compress or Route?" (Anonymous, 2026), we found”, “the first paper of this research series”, and “our prior work”, rather than treating it as neutral anonymous related work.

While the "Compress or Route" paper must be read by the Reviewer to get the background of the paper, doing such self-identification immediately reveals the identity of authors. This breaches double-blind practice.

I would suggest revising the paper before resubmission.

Best,